# Association between delirium and neutrophil percentage-to-albumin ratio in patients with cervical spinal cord injury: A single-center, retrospective study

Kosuke Nitta[1], Gentaro Kumagai[1]*, Kanichiro Wada[1], Yohshiro Nitobe[1], Kotaro Aburakawa[1], On Takeda[1], Kazushige Koyama[1], Hirotaka Kinoshita[2], Tetsuya Kushikata[2], Kazuyoshi Hirota[2,3], Yasuyuki Ishibashi[1]

1 Department of Orthopaedic Surgery, Hirosaki University Graduate School of Medicine, Zaifu-cho, Hirosaki, Aomori, Japan, 2 Department of Anesthesiology, Hirosaki University Graduate School of Medicine, Zaifu-cho, Hirosaki, Aomori, Japan, 3 Department of Perioperative Stress Management, Hirosaki University Graduate School of Medicine, Zaifu-cho, Hirosaki, Aomori, Japan

☉ These authors contributed equally to this work.
* gen722@hirosaki-u.ac.jp

## Abstract

### Study Design

Single-center, case-control study.

### Objective

This study investigated if the neutrophil percentage-to-albumin ratio (NPAR) could predict the onset of delirium in patients with cervical spinal cord injury (CSCI).

### Background

Delirium is a common and serious complication in patients with CSCI, leading to prolonged hospitalization and adverse clinical outcomes. Several risk factors have been identified, but the role of hematologic biomarkers in predicting delirium has remained unclear. While NPAR is a potential marker of systemic inflammation, its association with delirium in CSCI patients has not been established.

### Methods

The analysis included 147 patients with acute CSCI who were admitted to a single tertiary emergency center between 2010 and 2023. Delirium was diagnosed based on the Diagnostic and Statistical Manual of Mental Disorders, Fourth or Fifth Edition criteria. Clinical characteristics, laboratory data, and patient outcomes were compared between those with and without delirium. The association between NPAR and

**Data availability statement:** Data cannot be made publicly available due to ethical restrictions. The dataset contains human participant data with potentially identifying indirect identifiers (including precise age, sex, height, and weight), which could compromise participant privacy and the terms of consent. The minimal data set required to replicate the study's findings is available upon reasonable request from the Institutional Review Board of Hirosaki University Hospital (contact via [rinri@hirosaki-u.ac.jp]). Access will be granted to qualified researchers who meet the criteria for access to confidential data.

**Funding:** The author(s) received no specific funding for this work.

**Competing interests:** The authors have declared that no competing interests exist.

delirium was assessed using receiver operating characteristic (ROC) curve analysis and logistic regression models.

## Results

The incidence of delirium was 10.2% (15/147 patients). ROC analysis identified an NPAR cutoff value of 22 on the day of injury, with an AUC of 0.672. Multivariate logistic regression analysis revealed that an NPAR ≥22 (OR: 7.703, 95% CI: 2.151–27.584, $P$= .002) was independent risk factors for delirium.

## Conclusion

We found a significant association between a high NPAR (≥22) on the day of injury and the onset of delirium in CSCI patients. NPAR may be an accessible and effective biomarker for early delirium prediction, allowing timely interventions to improve patient outcomes.

## Introduction

Traumatic spinal cord injury (SCI) can be triggered by even minor trauma in older adults [1], and the prevalence of SCI is increasing as the population ages. A study by DeVivo et al. in the United States reported that the percentage of patients aged ≥60 years increased from 4.6% in 1970 to 20.4% in 2017 [2]. Miyakoshi et al. reported that patients aged ≥65 years comprised 66% of all SCI patients in Japan between 2010 and 2014 [1]. SCIs occurring in older adults are a major social problem, as they often lead to loss of independence and the need for long-term nursing care [3]. Therefore, the ability to predict and prevent complications and adverse events associated with SCI is especially important for older adults.

Delirium is defined as an acute, fluctuating disturbance of consciousness and cognitive function caused by physical illness or intoxication [4]. Delirium poses significant challenges for both patients and healthcare providers, contributing to prolonged hospital stays, delayed rehabilitation, a higher risk of complications, and increased demand for healthcare resources [5]. Cheung et al. reported that older age and lower motor scores on admission were associated with a higher risk of delirium in patients with SCI at any spinal level [6]. Currently, however, there are no established methods of predicting the onset of delirium in patients with limited cervical spinal cord injury (CSCI).

Several studies have investigated the association between the development of delirium and hematologic data in orthopedic diseases [7–9]. Neuroinflammation and oxidative stress are pathophysiologic mechanisms underlying the development of delirium after SCI [10]. The neutrophil percentage-to-albumin ratio (NPAR) has attracted attention as a simple inflammatory biomarker that can be obtained as a routine blood test item. Elevated NPAR is associated with poor prognosis in chronic renal failure [11] and cardiovascular disease [12]. The association between NPAR and the development of delirium in SCI patients, however, has remained unclear.

Based on our hypothesis that elevated NPAR levels are associated with neuroinflammation contributing to delirium, we investigated the association between NPAR and the development of delirium in patients with CSCI.

## Materials and methods

### Study design and subjects

This case-control study was conducted at a single tertiary emergency center in Japan, using medical records of acute CSCI patients who were admitted to and treated by our orthopedic department between 2010 and 2023. Spinal cord injuries were diagnosed by our orthopedic surgeons. A total of 156 patients with CSCI were included. After excluding patients with concomitant head trauma, 147 patients (116 males and 31 females) were divided into two groups: surgically treated and conservatively treated (Fig 1).

### Data collection

Patient characteristics such as age, sex, body mass index, injury levels, American Spinal Injury Association impairment scale (AIS), ossification of the posterior longitudinal ligament, diffuse idiopathic skeletal hyperostosis, diabetes, hypertension, association injuries, operation, alcohol use at the time of injury and post-hospitalization complications (delirium, pneumonia, urinary tract infection, and deep vein thrombosis) were collected. Alcohol use at the time of injury was collected as a categorical variable (yes/no) based on interviews with the patient or their family upon admission. In addition, laboratory data (white blood cell count [WBC], C-reactive protein [CRP] levels, neutrophil-to-lymphocyte ratio [NLR], and NPAR) were collected on the day of injury and on days 1 and 3 after injury. The blood test on the day of injury was a pre-operative sample collected upon hospital admission, while the samples on days 1 and 3 were collected on the morning of the respective following days. Although the blood data on days 1 and 3 post-injury included both surgical and conservatively treated patients, a sub-analysis confirmed that the performance of surgery did not significantly affect the values of inflammatory markers at these time points. The NLR was calculated as the absolute neutrophil count divided by the absolute lymphocyte count, and NPAR was calculated as (the percentage of neutrophils in total white blood cells [%] x100) divided by the blood albumin level (g/dL). For this retrospective analysis, we accessed the data necessary for our research purposes from June 12, 2024, to June 25, 2024. The dataset provided to the authors was anonymized, with all direct personal identifiers removed. The data included processed patient IDs to prevent duplicate records and enable data linkage.

### Definition and prevention of delirium

The primary outcome was the incidence of delirium. Complications were monitored until discharge. Delirium was defined based on the diagnostic criteria outlined in the Diagnostic and Statistical Manual of Mental Disorders, Fourth or Fifth

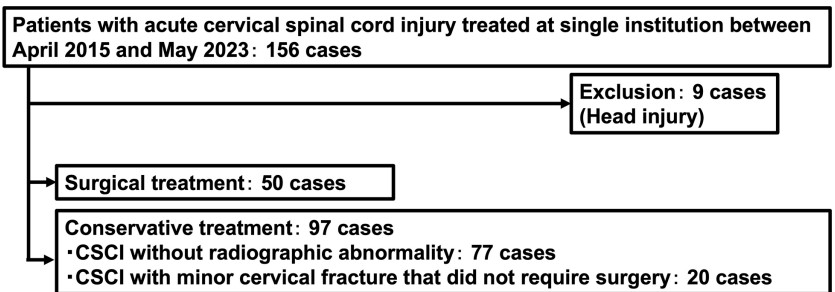

**Fig 1. Flow diagram of subject enrollment and inclusion in the study.** Patients with acute cervical spinal cord injury who were hospitalized and treated at Hirosaki University Hospital were categorized into cases treated surgically, cases with bone injuries, and cases with non-bone injury.

Edition. Immediately upon admission, nurses conducted a mental evaluation of all admitted patients, and contacted the attending psychiatrist if a sudden deterioration in mental status was detected. All patients suspected to have delirium based on the nurse's evaluation were referred to the psychiatry department by the attending physician. The incidence of delirium in patients with CSCI was calculated by counting the number of patients who developed delirium relative to the total number of patients with CSCI, with or without surgery. To prevent the development of delirium during hospitalization, nurses screened for risk factors for the development of delirium at the time of admission, and a policy of delirium control was implemented for patients deemed to be at high risk. According to our screening criteria [13], all patients admitted on an emergency basis for CSCI were classified as high-risk; therefore, the standardized prevention protocol was uniformly implemented for all 147 patients in this study cohort. A delirium management protocol, including measures such as avoiding the administration of non-benzodiazepine medications for insomnia, was implemented for high-risk patients to prevent the onset of delirium.

## Ethics statement

The study was approved by the Institutional Review Board of Hirosaki University Hospital (Approval Number: 2018−1002). This study is a retrospective observational study utilizing existing medical record data. Given that this study did not involve any interventions, human-derived samples, or direct contact with participants, and posed minimal risks, the Institutional Review Board determined that it met the criteria for a waiver of informed consent under the "Ethical Guidelines for Medical Research Involving Human Subjects." Participants' right to refuse participation, to access and request modifications to their data, and the assurance of data transfer between institutions will be communicated and made publicly available on the Hirosaki University Hospital website and other appropriate means. Participants retained the right to withdraw their data from the study or request amendments to their registered information at any time without any repercussions.

## Statistical analysis

To explore patient characteristics associated with delirium, we performed statistical comparisons between patients with and without delirium, using the Mann-Whitney U test for continuous variables and the chi-square test for categorical variables. Receiver operating characteristic (ROC) curve analysis was performed to evaluate the association between the blood test items selected as dependent variables and the onset of delirium. We established cutoff values for WBC, CRP, NLR, and NPAR. The optimal cutoff value was determined using the Youden index ($J = sensitivity + specificity − 1$), which maximizes the sum of sensitivity and specificity. Univariate analyses followed by multivariate logistic regression were performed to determine independent predictors of delirium following CSCI. Age, sex, body mass index, medical history, severity of paralysis (AIS A or B), and cutoff values (NPAR, on the day of injury) were used as the independent variables. Independent variables in the multivariate analysis included factors with $P$-values <0.1 after univariate analysis. The Statistical Package for the Social Sciences (SPSS Inc, Chicago, IL, USA) version 29.0 was used for the statistical analysis. The threshold of statistical significance was set at $P < .05$. Power analysis was performed using G*Power 3.1, with the Mann-Whitney U test assuming a logistic parent distribution. The calculated power was 0.762, which is slightly below the typical target of 0.80. However, this value is considered acceptable within the design of the study.

## Results

Of the 147 patients included in the analysis, 15 (10.2%; 11 [11.3%] of whom underwent conservative treatment and 4 [8.0%] of whom underwent surgery) developed delirium after CSCI (Fig 2). In the delirium group, the diagnosis was made at a mean of 2.9 days (range: 2–5 days) post-injury. The characteristics of patients who developed delirium and those who did not were compared. Patients diagnosed with delirium had a significantly higher frequency of alcohol use at the time of injury ($P = .040$) and of pneumonia after injury ($P = .005$; Tables 1 and 2). Among the laboratory data, NPAR on the day of injury, the primary focus of this study, was significantly higher in the delirium group ($P = .029$; Table 2). To investigate the basis for this elevation,

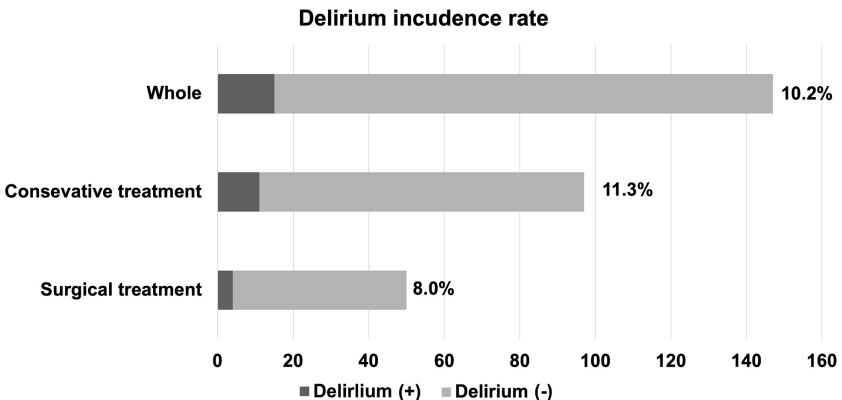

**Delirium incidence rate**

Whole — 10.2%
Consevative treatment — 11.3%
Surgical treatment — 8.0%

■ Delirlium (+)　■ Delirium (-)

**Fig 2. Incidence of delirium in the present study.** Patients with acute cervical spinal cord injury (n = 147) were categorized as conservatively treated or surgically treated, and the incidence of delirium was 10.2%, 11.3%, and 8.0%, respectively.

we analyzed its components and found that the delirium group had a trend toward significantly lower serum albumin levels (*P* = .061), while there was no significant difference in absolute neutrophil counts between the groups.

We used ROC curves to determine the NPAR cutoff value associated with the onset of delirium after CSCI. We performed ROC analysis on the day of injury, day 1 post-injury, and day 3 post-injury (Fig 3 and Table 3). NPAR had the highest area under the ROC curve on the day of injury (AUC: 0.672; 95% CI: 0.503-0.840), with a cutoff value of 22 (Table 3). The sensitivity was 66.7% and the specificity was 18.2%. Univariate logistic analysis revealed that alcohol use at the time of injury (*P* = .037) and NPAR >22 on the day of injury were significantly associated with the development of delirium after

**Table 1. Demographic data of patients with cervical spinal cord injury(n = 147).**

|  | Delirium (+) (n = 15) | Delirium (-) (n = 132) | *P*-value[b] |
|---|---|---|---|
| Age[a] | 72.0 [67.0,77.0] | 69.0 [60.0,77.0] | 0.321 |
| Male, n, (%) | 13 (86.7) | 103 (78.4) | 0.346 |
| BMI[a] (kg/m²) | 22.9 [19.4,26.0] | 22.6 [20.6,24.2] | 0.974 |
| OPLL, n, (%) | 1 (6.7) | 4 (3.0) | 0.421 |
| DISH, n, (%) | 4 (26.7) | 12 (9.1) | 0.061 |
| AIS AB, n, (%) | 4 (26.7) | 28 (21.2) | 0.419 |
| DM, n, (%) | 4 (26.7) | 28 (21.2) | 0.419 |
| Hypertension, n, (%) | 5 (33.3) | 49 (37.1) | 0.506 |
| Association injuries n, (%) | 5 (33.3) | 18 (13.6) | 0.061 |
| Operation, n, (%) | 4 (26.7) | 46 (34.8) | 0.374 |
| Alcohol use at the time of injury, n, (%) | 6 (40.0) | 22 (18.1) | 0.040* |
| Complication |  |  |  |
| DVT, n, (%) | 5 (33.3) | 15 (11.4) | 0.034* |
| Pneumonia, n, (%) | 7 (46.7) | 18 (13.6) | 0.005* |
| Urinary infection, n, (%) | 3 (20.0) | 10 (7.6) | 0.131 |

BMI, body mass index; OPLL, ossification of posterior longitudinal ligament; DISH, Diffuse idiopathic skeletal hyperostosis; AIS, the American Spinal Cord Injury Association Impairment; DM, diabetes mellitus; DVT, Deep vein thrombosis

[a] Values are presented as median and quartiles

[b] Significant differences (P < 0.05) between values for patients and without delirium were calculated using the *Mann-Whitney U test, #chi-square test

**Table 2. Laboratory data in patients with CSCI (n = 147).**

| | Delirium (+) (n = 15) | Delirium (-) (n = 132) | P-value[b] |
|---|---|---|---|
| Age[a] | 72.0 [67.0,77.0] | 69.0 [60.0,77.0] | 0.321 |
| Men, n, (%) | 13 (86.7) | 103 (78.4) | 0.277 |
| BMI[a] (kg/m$^2$) | 22.9 [19.4,26.0] | 22.6 [20.6,24.2] | 0.974 |
| AIS AB, n, (%) | 4 (26.7) | 28 (21.2) | 0.419 |
| Operation, n, (%) | 4 (26.7) | 46 (34.8) | 0.374 |
| Alcohol use at the time of injury, n, (%) | 6 (40.0) | 22 (18.1) | 0.040# |
| WBC (10^3/μL)[a] | | | |
| injured day | 6.9 [4.9,12.3] | 8.0 [6.2,10.4] | 0.650 |
| 1 day after injury | 8.3 [6.0,10.1] | 8.5 [7.2,10.1] | 0.474 |
| 3 days after injury | 8.8 [6.0,11.6] | 8.2 [6.2,10.1] | 0.850 |
| Absolute Neutrophil count (10^3/μL)[a] | | | |
| injured day | 4.6 [3.8,10.8] | 6.0 [4.2,8.1] | 1.000 |
| 1 day after injury | 6.7 [4.6,8.5] | 6.6 [5.2,8.0] | 0.582 |
| 3 days after injury | 6.3 [4.6,9.4] | 6.1 [4.5,8.1] | 0.506 |
| Alb (g/dL)[a] | | | |
| injured day | 3.5 [3.3,4.0] | 3.9 [3.6,4.1] | 0.061 |
| 1 day after injury | 3.1 [2.9,3.3] | 3.3 [2.9,3.6] | 0.143 |
| 3 days after injury | 2.8 [2.7,3.2] | 3.1 [2.7,3.5] | 0.221 |
| NLR[a] | | | |
| injured day | 6.8 [3.2,14.1] | 4.4 [2.6,6.8] | 0.089 |
| 1 day after injury | 5.5 [4.3,8.0] | 6.1 [4.2,8.3] | 0.878 |
| 3 days after injury | 6.0 [4.1,10.0] | 5.6 [3.4,7.8] | 0.256 |
| CRP (mg/dL)[a] | | | |
| injured day | 0.3 [0.1,1.1] | 0.1 [0.0,0.5] | 0.073 |
| 1 day after injury | 2.2 [0.6,4.9] | 1.9 [0.4,4.4] | 0.636 |
| 3 days after injury | 3.3 [2.3,12.1] | 4.6 [0.9,9.9] | 0.715 |
| NPAR[a] | | | |
| day of injury | 22.8 [18.0,23.6] | 19.5 [16.4,21.7] | 0.029* |
| 1 day after injury | 24.5 [22.1,28.4] | 23.4 [21.0,26.5] | 0.255 |
| 3 days after injury | 27.2 [22.7,30.4] | 24.9 [20.9,27.9] | 0.096 |

BMI, body mass index; AIS, the American Spinal Cord Injury Association Impairment; WBC, white blood cells; Alb, Serum Albumin; NLR, neutrophil-lymphocyte ratio; NPAR, neutrophil percentage-to-albumin ratio CRP, C-Reactive Protein.

[a] Values are presented as median and quartiles

[b] Significant differences (P < 0.05) between values for patients and without delirium were calculated using the *Mann-Whitney U test, #chi-square test

injury. Multivariate logistic regression analysis showed that NPAR >22 (odds ratio: 7.703, 95% CI: 2.151-27.584, P =.002) on the day of injury was significantly associated with the onset of delirium (Table 4).

## Discussion

The present study investigated the incidence of delirium and associated factors in a population of patients with CSCI. In patients with CSCI, the incidence of delirium was 10.2%. Univariate logistic regression analysis identified alcohol use at the time of injury as a significant risk factor for delirium. Multivariate logistic regression and ROC

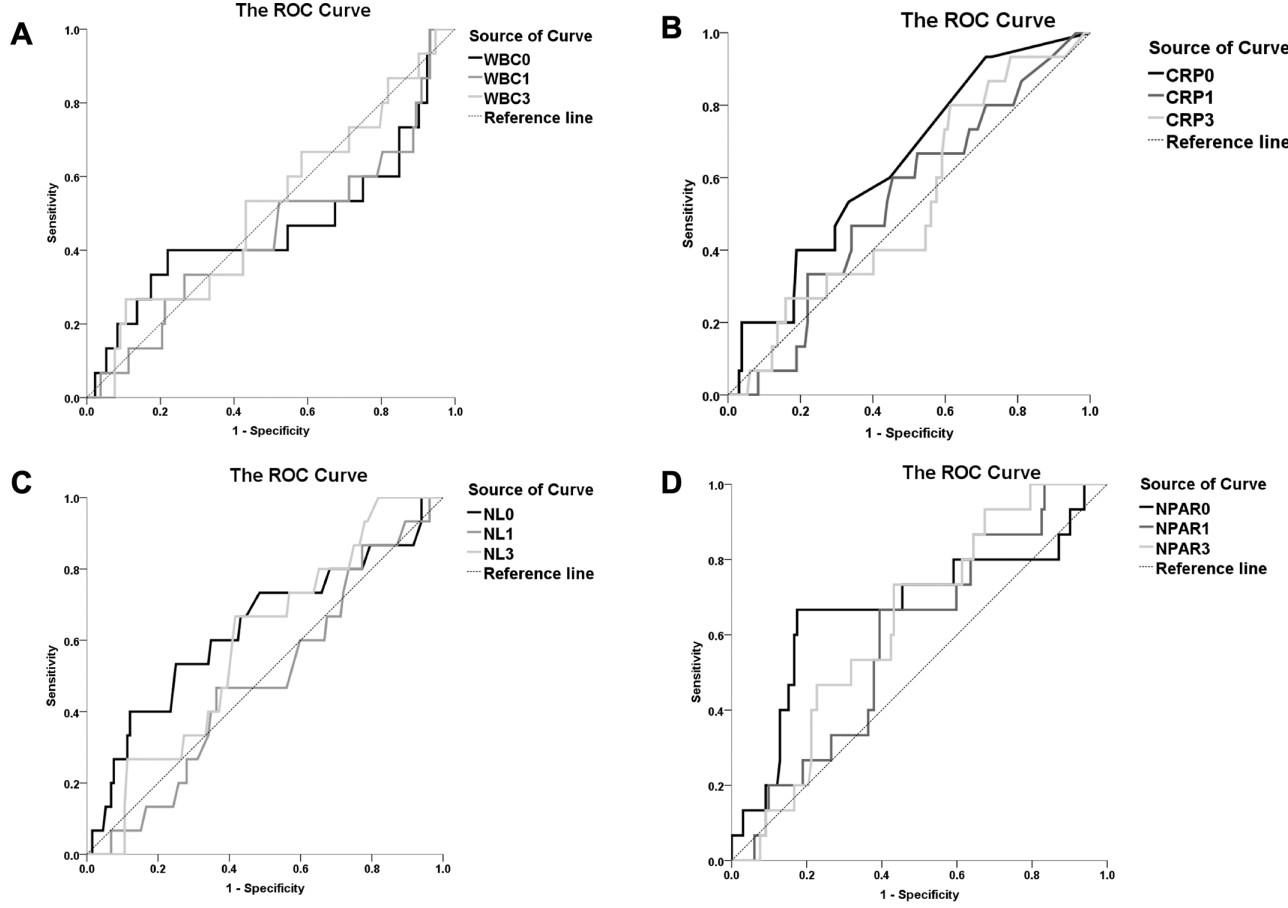

**Fig 3. ROC curve of WBC (A), CRP (B), NLR (C), NPAR (D) on the day of spinal cord injury, 1 day after injury, and 3 days after injury.**

curve analyses demonstrated a significant association between NPAR >22 at the time of injury and the development of delirium.

The incidence of postoperative delirium after elective spinal surgery in our department is 9.2% [13], which is almost the same rate as in the present study (10.2%). Street et al. [14] and Cheung et al. [15] reported delirium incidences of 18.7% and 17.7%, respectively, in patients with traumatic SCI. Compared to prior studies, the lower incidence of delirium after SCI in this study may stem from variations in the diagnostic criteria. In addition, some high-risk patients in the present study were prescribed sleeping pills in advance, which may have prevented the onset of delirium, contributing to the lower incidence rate.

Tamai et al. reported that six items for scoring the risk of developing delirium in patients with CSCI: older age, hypoalbuminemia, cervical spine fracture, organ damage, activity before the injury, and diabetes [16]. Some reports indicate that the use of carbapenems [17] and vitamin B12 deficiency [18] contributes to the development of delirium in patients with traumatic SCI. In the present study, surgical intervention was not significantly associated with the development of delirium. Consistent with findings from previous studies [16,19], we also did not identify a significant association between the severity of paralysis and the development of delirium. In our study, older age was not a statistically significant factor for delirium. Given the small number of patients in the delirium group (n=15), it is possible that our study was underpowered to detect a significant association. Considering that older age is a well-established risk factor for delirium in previous literature [16],

**Table 3. ROC curve analysis (n = 147).**

|  | AUC | cut-off value | p-value | 95%CI |
|---|---|---|---|---|
| WBC |  |  |  |  |
| injured day | 0.464 | 10.9 | 0.650 | 0.278-0.651 |
| 1 day after injury | 0.457 | 11.1 | 0.474 | 0.276-0.611 |
| 3 days after injury | 0.516 | 8.8 | 0.850 | 0.356-0.674 |
| CRP |  |  |  |  |
| injured day | 0.65 | 0.4 | 0.079 | 0.501-0.776 |
| 1 day after injury | 0.599 | 2.0 | 0.636 | 0.395-0.680 |
| 3 days after injury | 0.53 | 2.3 | 0.715 | 0.386-0.672 |
| NLR |  |  |  |  |
| injured day | 0.634 | 5.7 | 0.089 | 0.470-0.799 |
| 1 day after injury | 0.488 | 7.4 | 0.878 | 0.343-0.633 |
| 3 days after injury | 0.590 | 6.0 | 0.256 | 0.455-0.724 |
| NPAR |  |  |  |  |
| injured day | 0.672 | 22 | 0.029 | 0.503-0.840 |
| 1 day after injury | 0.590 | 24 | 0.255 | 0.454-0.725 |
| 3 days after injury | 0.632 | 25 | 0.096 | 0.506-0.757 |

AUC, Area under the curve; WBC, white blood cells; NLR, neutrophil-to-lymphocyte ratio; CRP, C-reactive protein; NPAR, neutrophil percentage-to-albumin ratio.

**Table 4. Univariate and multivariate logistic regression analysis (n = 147).**

|  | Univariate regression analysis | | | Multivariate regression analysis | | |
|---|---|---|---|---|---|---|
|  | OR | 95%CI | *P* value | OR | 95%CI | *P* value |
| Age | 1.026 | 0.979-1.075 | 0.286 | 1.031 | 0.965-1.101 | 0.367 |
| Men, n, (%) | 0.546 | 0.117-2.561 | 0.443 | 0.560 | 0.100-3.128 | 0.508 |
| BMI (kg/m$^2$) | 1.022 | 0.878-1.189 | 0.780 | 1.074 | 0.906-1.273 | 0.410 |
| AIS AB | 1.351 | 0.400-4.566 | 0.629 | 0.853 | 0.204-3.572 | 0.827 |
| Operation | 0.680 | 0.205-2.255. | 0.629 | 0.557 | 0.150-2.075 | 0.384 |
| Alcohol use at the time of injury, n, (%) | 3.333 | 1.077-10.318 | 0.037* | 3.462 | 0.813-14.748 | 0.093 |
| NPAR≥22 at injured day | 7.931 | 2.348-23.502 | <0.001* | 7.703 | 2.151-27.584 | 0.002* |

Results of multiple regression analysis for prediction of delirium; OR, odds ratio; BMI, body mass index; CI, confidence interval; NPAR, Neutrophil Percentage-to-Albumin ratio. Significant differences (*P* < .05) between values for patients and those without delirium were calculated using univariate and multivariate regression analyses.

this association may become significant in a larger cohort study. Previous reports suggested that older adults are more likely to develop low-activity delirium as a subtype of delirium [20], and it is possible that we missed cases of low-activity delirium in older patients.

In the present study, NPAR on the day of injury was associated with the development of delirium following CSCI. Recently, NPAR has emerged as a novel predictor of systemic inflammation and infection, and is reportedly associated with all-cause mortality in patients with coronary artery disease, myocardial infarction, cardiogenic shock, sepsis, septic shock, and acute kidney injury [11,12,21–24]. An association between NPAR and depression has also been reported [25]. In the context of SCI, Wang et al. reported that an NPAR cutoff value of 21.7 was associated with the development of pneumonia [26], closely aligning with the cutoff value determined in our study. In their report, they suggested that the high

rate of pneumonia observed in SCI patients in their study was due to a large number of neutrophils in the lungs, increased NPAR, and the production of inflammatory cytokines and cytokine factors [26].

NPAR may also reflect underlying mechanisms related to the development of delirium. It is important to clarify that NPAR is a marker of systemic inflammation, and our study possesses no direct evidence of neuroinflammation or blood-brain barrier disruption. Therefore, our hypothesis regarding the mechanism remains speculative. However, recent studies have reported associations between elevated NPAR and other central nervous system disorders such as depression and stroke [25, 27, 28]. Given these findings, it is plausible to hypothesize that the severe systemic inflammation indicated by a high NPAR may contribute to neuroinflammatory processes following the initial spinal cord injury. The precise mechanisms underlying this potential link warrant future investigation. Maldonado et al. suggested that inflammatory cytokines stimulated by infection, surgical procedures, or trauma can activate proinflammatory cytokines in the brain, potentially worsening age-related neuroinflammation in the frontal and temporal lobes [29]. Hight et al. further suggested that central nervous system inflammation may induce delirium by slowing EEG activity, especially alpha waves [30]. We hypothesized that temporary disruption of the cerebrospinal barrier due to SCI may allow an influx of inflammatory cytokines into the cerebrospinal cord, thereby increasing susceptibility to delirium after SCI. Albumin, a key component of the NPAR calculation, has neuroprotective effects through its ability to mitigate neuroinflammation and oxidative stress [31]. Therefore, as suggested by Zawiah et al [32], NPAR may be a more clinically relevant indicator than NLR for assessing central nervous system inflammation. NPAR integrates both inflammatory insult (neutrophils) and neuroprotective capacity (albumin), making it a more mechanistically relevant biomarker for neuroinflammation in SCI than non-specific inflammatory markers, such as CRP and WBC. In our cohort, the NPAR cutoff value of 22 identified for predicting delirium was substantially higher than the reported mean of 13.72 ± 2.65 in a large cohort of 33,768 non-trauma adults [25], suggesting a pronounced state of physiological stress. Notably, the elevated NPAR in the delirium group appeared to be driven primarily by lower serum albumin levels rather than by an increase in neutrophil counts. Given that serum albumin reflects not only nutritional status but also possesses antioxidant and neuroprotective properties, a lower albumin level at the time of injury may indicate an increased vulnerability to delirium development. Thus, NPAR may serve as a more comprehensive biomarker that integrates both inflammatory activity and the patient's physiological resilience.

Although the exact relationship between inflammatory responses and the development of delirium remains unclear, our findings support the potential of NPAR as a novel predictor of delirium in SCI patients. Further large-scale prospective studies and experimental models are necessary to clarify the underlying pathophysiology and validate these associations. For our multivariate model, we included alcohol use at the time of injury as it is a well-established risk factor for delirium. Conversely, we excluded subsequent complications like pneumonia. The critical rationale for this was temporal: in our cohort, all cases of pneumonia were diagnosed only after the onset of delirium, meaning it could not be a predictor. Unlike these later complications, NPAR measured upon admission holds significant clinical value as a predictive tool that can facilitate early intervention.

In this study, alcohol use at the time of injury was significantly associated with the development of delirium. Yokota et al. reported that 16.1% of patients with traumatic SCI had fallen due to alcohol use at the time of injury [33]. They also noted that older adults are more likely to suffer SCI due to alcohol use and are more frequently affected at higher spinal levels. Earlier studies also cited heavy alcohol use as a risk factor for postoperative delirium following lumbar spine surgery [34]. Although 19.0% of patients in the present study had used alcohol at the time of injury, the proportion of patients was significantly higher among those who developed delirium compared with those who did not (40.0% vs. 18.2%, *P* =.040). Collecting information from patients' families regarding habitual alcohol use and the circumstances of the injury may help predict delirium onset and implement preventive countermeasures. Admittedly, NPAR is a non-specific marker of systemic inflammation, and more specific cytokines like IL-6 have been linked to delirium [35]. The primary strength of NPAR, however, lies in its clinical utility: it can be calculated simply and rapidly from routine blood tests in the emergency setting. Our study focused specifically on the day-of-injury NPAR because it provides the earliest opportunity for prediction, free

from potential confounders such as surgical intervention. Unlike a post-onset complication such as pneumonia, NPAR assessed at admission is a valuable and pragmatic tool for identifying high-risk patients early and initiating preventive care.

This study has several limitations. First, it was conducted at a single center, which may limit the generalizability of our findings. Second, due its retrospective design, causal relationships cannot be determined. We state that future validation through large-scale, prospective, multicenter studies is essential. Third, instead of treating NPAR as a continuous SCI variable, we used an NPAR cut-off value of ≥22 as an independent factor in the logistic analysis. Our cut-off value was consistent with that in previous studies evaluating the relationship between SCI and NPAR [26]. We propose that the next logical step for this research is to move beyond a single biomarker. We suggest a future direction of developing a multi-factorial risk scoring system, which would incorporate NPAR with other relevant clinical factors to create a more robust and accurate tool for predicting delirium. Fourth, not all patients with SCI received a psychiatric consultation, and we may have missed cases of low-activity delirium. Fifth, the assessment of alcohol use relied on patient or family reporting rather than objective measures such as blood alcohol concentration and thus may be subject to reporting bias. Finally, we did not evaluate cognitive decline using the Mini-Mental State Examination or assess delirium severity with a Confusion Assessment Method-based scoring system [36].

## Conclusion

In this single-center, case-control study, the incidence of delirium in patients with CSCI was 10.2%. Elevated NPAR was associated with an increased risk of delirium. NPAR may serve as a simple and measurable biomarker for predicting delirium in patients following CSCI.

## Acknowledgments

We thank SciTechEdit (https://scitechedit.com/) for assistance with English language editing.

## Author contributions

**Data curation:** Kosuke Nitta.

**Formal analysis:** Kosuke Nitta.

**Investigation:** Kosuke Nitta.

**Visualization:** Kosuke Nitta.

**Writing – original draft:** Gentaro Kumagai, Kosuke Nitta.

**Writing – review & editing:** Gentaro Kumagai, Kanichiro Wada, Yohshiro Nitobe, Kotaro Aburakawa, On Takeda, Kazushige Koyama, Hirotaka Kinoshita, Tetsuya Kushikata, Kazuyoshi Hirota, Yasuyuki Ishibashi.

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
