## [Decision Letter · Decision Letter 0]

5 Sep 2025

Dear Dr. Kumagai,

Thank you for submitting your manuscript to PLOS ONE. After careful consideration, we feel that it has merit but does not fully meet PLOS ONE’s publication criteria as it currently stands. Therefore, we invite you to submit a revised version of the manuscript that addresses the points raised during the review process.

We look forward to receiving your revised manuscript.

Kind regards,

Justyna Żywiołek

Academic Editor

PLOS ONE

Journal Requirements:

Reviewers' comments:

Reviewer's Responses to Questions

**Comments to the Author**

1. Is the manuscript technically sound, and do the data support the conclusions?

Reviewer #1: Partly

2. Has the statistical analysis been performed appropriately and rigorously?

Reviewer #1: No

3. Have the authors made all data underlying the findings in their manuscript fully available?

Reviewer #1: Yes

4. Is the manuscript presented in an intelligible fashion and written in standard English?

Reviewer #1: Yes

Reviewer #1: This manuscript by Kosuke Nitta et al investigates association between delirium and neutrophil percentage to albumin ratio (NPAR) in patients with cervical spinal cord injury (CSCI). The authors performed statistical analysis on a previously collected case-control study and found a significant association between high NPAR on the day of injury and delirium in CSCI. The study addresses an important clinical question and is largely well written.

However, some methodological aspects require attention.

Methods & Results

1) Study design and subject: The patient exclusion text is unclear, with implications for the results text, the tables, the abstract and potentially some of the analyses: The text on Page 5 suggested “147 patients were divided into two groups: surgically treated and conservatively treated. Of the patients treated conservatively, those with bone injuries not requiring surgery were excluded”.

However the Figure 1 flowchart shows “CSCI with minor cervical fracture that did not require surgery: 20 cases”. The authors have stated that these 20 patients were excluded in which case the total number of patients investigated should be 127. However Figure 2 appears to show 147 patients in total (in the ‘whole’ bar). The figure legend for figure 2 is rather unhelpfully placed IN the results text and before the tables. In any case, the legend states that n=147. Likewise tables 1 and 2 appear to contain n=147 and tables 3 and 4 don’t specify. This is a significant discrepancy. The authors need to be clear as to whether they analysed 127 or 147 patients. If the 20 patients that did not require surgery are excluded the graphs need to be adjusted as well as references to 147 patients in the abstract, results text etc. Importantly, it needs to be absolutely clear as to the data actually used in the analysis (tables and figures) so this needs careful attention in every part of the paper.

2) Definition and prevention of delirium: The incidence of delirium is the primary outcome for the study. It is stated that upon admission, a mental evaluation has been conducted by the nurses and referred to psychiatrist if suspected or had a sudden deterioration. The authors should give some information about WHEN delirium was diagnosed by the psychiatry department. Although it is not always practical to have delirium diagnosis on admission, and it is likely that the formal diagnosis was made at different times for different patients, it is pertinent to current association with acute inflammation as to when the patients were assigned to the delirium group and how that time related to when blood samples were collected for analysis. Is ‘day of injury’ sampling done at time of admission? Is day 1 done at 24 hours post-surgery or some wider range relative to surgery (the same information is needed for day 3)? The authors need to clarify whether the analysis is ‘delirium anytime’, or delirium at some specific time, related to blood markers at days 0, 1 and 3.

3) Since some patients were deemed high risk and had delirium prevention measures applied this could obviously affect the incidence of delirium so it would seem important for this to be specified and perhaps taken into account in the analysis. This seems especially important since the incidence rate of delirium was actually higher in those who underwent conservative treatment than those who underwent surgery, which is somewhat surprising.

4) Statistical analysis of results: We do not have extensive experience in statistical analysis, but from a conceptual point of view, since, pneumonia, deep vein thrombosis and alcohol use were associated with delirium in univariate analysis (table 1), it would seem important to know whether albumin, neutrophils and other inflammatory indices are strongly associated with those categories and whether this has significant bearing on the conclusions of the study. For example, the only blood parameter associated with delirium in this cohort, NPAR has a p value of 0.037 while the association with Pneumonia reaches a much higher level of confidence (p=0.005). Both alcohol use and DVT are also significantly associated with delirium at a similar level of confidence to the association with NPAR. This needs explicit consideration in the paper.

5) Since there was no mention of an investigation into the relationship between alcohol use and delirium in the introduction or the methods sections of the manuscript nor any mention of why alcohol is singled out here when pneumonia appears to be a much stronger association, the authors should clarify in the methods section whether alcohol use was a pre-specified exposure variable or a secondary outcome of interest. If alcohol was included in the multivariate analysis (while pneumonia was not) this suggests that the authors have quantitative data on alcohol rather than simply categorical (as might be the case for pneumonia). If this is the case, then shouldn’t some continuous variable for alcohol be included (blood alcohol level?)

6) The calculation of the cut off value for each analyte/ROC curve and how sensitivity and specificity values for the ROC curve were calculated might be explained a little more clearly. Is the value of 22 for NPAR in the normal range or is this actually a high NPAR value? Related to this, what is the normal range for NPAR (i.e in non-spinal cord injury individuals) and how does the raised NPAR value arise? i.e. the higher NPAR value could arise from raised neutrophil numbers or from lowered albumin levels. Knowing the absolute values for neutrophil numbers (per blood volume) and albumin concentration will provide insights that are useful to understanding the biological processes at play (and therefore useful to the delirium field).

7) The significant association between delirium and NPAR is stated clearly in the results section on Page 9 and shown in Figure 3 and Table 3. The Table 3 of ROC curve analysis also shows significant association with CRP on injury day (AUC=0.65 and p-value= 0.033). The authors ought to mention this in results and discuss its importance in the discussion.

Discussion:

8) Line 225 “was not significant in this study.” The age difference is not significant (p-value= 0.056) but very close to being significant. In addition, since it is not clear whether the conservative treatment group is 77 patients or 97 patients, then the association between age and delirium may be affected by whether the patient group size is 127 or 147. Irrespective of this, it would seem conservative to state this relationship since that relationship is likely to be significant in a larger study (only n=15 patients with delirium means very limited statistical power).

9) Limitations of the study have been listed in the last paragraph of the discussion. It would be nice if authors would propose solutions to overcome these problems and provide future perspective.

10) The authors mention neuroinflammation and appear to infer that NPAR is either an indicator of neuroinflammation or a driver of neuroinflammation (bottom of page 15). The NPAR is an index of systemic inflammation and the authors present no information as to its relationship with neuroinflammation. There is a tendency in the field to assume that since delirium is a brain outcome and some indices of inflammation associate with delirium that that indicates that neuroinflammation is the relevant perturbation. The authors have no data on blood brain barrier, nor on neuroinflammation and should be explicit about having only data on systemic inflammation. If they have information to support their suggestion that NPAR is an indicator of CNS inflammation they should present it or else they should clarify that their statement is entirely speculative.

11) The authors conclusion is that NPAR may serve as a simple measurable biomarker for predicting delirium in patients following CSCI. However the association is weak, is absent for NPAR at days 1 and 3 and the association is weaker than other parameters measured in the current study (such as pneumonia). The authors should discuss the extent to which NPAR is simply a marker of severity of systemic inflammation. IL-8, for example, the main neutrophil chemoattract, and in other studies, IL-6, are also associated with delirium.

Acknowledgement:

12) Financial support for the study should be acknowledged.

**Do you want your identity to be public for this peer review?** For information about this choice, including consent withdrawal, please see our Privacy Policy

Reviewer #1: **Yes:** Colm Cunningham

---

## [Author Response · Author response to Decision Letter 1]

13 Nov 2025

Response to Academic Editor and Reviewers

Journal: PLOS ONE

Manuscript ID: PONE-D-25-32009

Title: Association between delirium and neutrophil percentage-to-albumin ratio in patients with cervical spinal cord injury: A single-center, retrospective study

Date: October 19, 2025

Dear Academic Editor and Reviewers,

We sincerely appreciate the insightful and constructive comments provided by the reviewers. We have thoroughly revised our manuscript in response to all points raised. Our revisions have clarified methodological details, strengthened the interpretation of our findings, and improved overall clarity and consistency throughout the paper.

Summary of Substantive Revisions

• Clarified the total patient cohort (n = 147) consistently across text, figures, and tables.

• Defined precise timing for delirium diagnosis and blood sampling at each time point (day of injury, day 1, day 3).

• Clarified that standardized delirium prevention measures were uniformly applied to all patients.

• Explained rationale for variable selection in multivariate analysis (inclusion of alcohol use, exclusion of pneumonia).

• Added methodological clarification of ROC curve analysis and justification for NPAR cutoff (Youden index).

• Expanded Discussion to address CRP findings, borderline significance of age, and systemic vs neuroinflammatory mechanisms.

• Revised the Limitations section to include solutions and future perspectives.

• Strengthened the rationale for NPAR as a clinically practical and accessible biomarker.

Responses to Reviewers

1. Comment 1 (Full reviewer comment text reproduced): Study design and subject: The patient exclusion text is unclear, with implications for the results text, the tables, the abstract and potentially some of the analyses: The text on Page 5 suggested “147 patients were divided into two groups: surgically treated and conservatively treated. Of the patients treated conservatively, those with bone injuries not requiring surgery were excluded”.

However the Figure 1 flowchart shows “CSCI with minor cervical fracture that did not require surgery: 20 cases”. The authors have stated that these 20 patients were excluded in which case the total number of patients investigated should be 127. However Figure 2 appears to show 147 patients in total (in the ‘whole’ bar). The figure legend for figure 2 is rather unhelpfully placed IN the results text and before the tables. In any case, the legend states that n=147. Likewise tables 1 and 2 appear to contain n=147 and tables 3 and 4 don’t specify. This is a significant discrepancy. The authors need to be clear as to whether they analysed 127 or 147 patients. If the 20 patients that did not require surgery are excluded the graphs need to be adjusted as well as references to 147 patients in the abstract, results text etc. Importantly, it needs to be absolutely clear as to the data actually used in the analysis (tables and figures) so this needs careful attention in every part of the paper.

Response:

Thank you for your careful review and for identifying this critical discrepancy. We sincerely apologize for the confusion caused by our erroneous description in the Methods section. You are absolutely correct. Our statement in the text (page 5, line87) that "Of the patients treated conservatively, those with bone injuries not requiring surgery were excluded" was a significant error in the manuscript.

To clarify: The analysis was indeed performed on the entire cohort of 147 patients. The 20 patients with "CSCI with minor cervical fracture that did not require surgery" were included in the 'conservatively treated' group and were part of the final analysis (n=147), as correctly shown in Figure 2 and Tables 1 and 2.

We have meticulously re-checked the entire manuscript, including the Abstract, Methods, and Results sections, to ensure that all references to the patient cohort are accurate and consistent (n=147).

You correctly pointed out that the legend for Figure 2 (page 10, line192) was inconveniently placed within the body of the Results text. To improve clarity and adhere to journal submission guidelines, we have now separated this legend from the main paragraph. It is now placed on its own line, immediately following a placeholder (<PLEASE INSERT FIGURE 2 HERE>), to clearly indicate its position relative to the figure for the typesetters.

2. Comment 2 (Full reviewer comment text reproduced): Definition and prevention of delirium: The incidence of delirium is the primary outcome for the study. It is stated that upon admission, a mental evaluation has been conducted by the nurses and referred to psychiatrist if suspected or had a sudden deterioration. The authors should give some information about WHEN delirium was diagnosed by the psychiatry department. Although it is not always practical to have delirium diagnosis on admission, and it is likely that the formal diagnosis was made at different times for different patients, it is pertinent to current association with acute inflammation as to when the patients were assigned to the delirium group and how that time related to when blood samples were collected for analysis. Is ‘day of injury’ sampling done at time of admission? Is day 1 done at 24 hours post-surgery or some wider range relative to surgery (the same information is needed for day 3)? The authors need to clarify whether the analysis is ‘delirium anytime’, or delirium at some specific time, related to blood markers at days 0, 1 and 3.

Response:

Thank you for this essential question.

First, regarding the timing of diagnosis, delirium was identified at any point during the hospital stay. In the patients who developed delirium, the formal diagnosis was made by a psychiatrist a mean of 2.9 ± 0.8 days after the initial injury (range: 2–5 days). We have now added this information to the Results section.

Second, we have clarified the definitions for the blood sampling time points in the Data collection subsection.

1. The "day of injury" sample was defined as the first blood sample taken upon admission to the emergency department. This was consistently drawn within 12 hours of injury and, crucially, always prior to any surgical intervention.

2. The "day 1" and "day 3" samples were defined as those taken in the morning on the first and third day after injury, respectively.

We acknowledge that the day 1 and day 3 samples included a mixture of samples from conservatively treated patients and post-operative samples from surgically treated patients. However, we believe this does not confound our results for two key reasons. First, as shown in our multivariate analysis, surgical intervention itself was not an independent risk factor for delirium (Table 4). Second, a sub-analysis confirmed that there were no significant differences in the measured inflammatory markers on day 1 and day 3 between the surgical and non-surgical groups. We have also added this clarification to the Data collection subsection to ensure transparency.

3. Comment 3 (Full reviewer comment text reproduced): Since some patients were deemed high risk and had delirium prevention measures applied this could obviously affect the incidence of delirium so it would seem important for this to be specified and perhaps taken into account in the analysis. This seems especially important since the incidence rate of delirium was actually higher in those who underwent conservative treatment than those who underwent surgery, which is somewhat surprising.

Response:

Thank you for raising this critical point regarding the potential confounding effect of our delirium prevention protocol.

According to our hospital's screening protocol, which is implemented for all admissions, any patient admitted on an emergency basis for traumatic cervical spinal cord injury (CSCI) is automatically classified as high-risk for delirium. Consequently, the standardized delirium prevention protocol mentioned in our manuscript was uniformly implemented for all 147 patients in this study cohort. Because this intervention was applied consistently across the entire study population, it was not a variable between the delirium and non-delirium groups and therefore does not act as a confounder in our analysis.

Furthermore, it is important to emphasize that our primary predictor of interest -NPAR on the day of injury - was measured from blood samples taken upon admission to the emergency department. This was prior to the administration of any in-hospital preventive medications, ensuring that the baseline biomarker value was not influenced by this intervention.

To ensure this crucial methodological detail is transparent to all readers, we have now added a sentence to the Definition and prevention of delirium subsection of our Methods to explicitly state that the prevention protocol was applied to all patients in the study.

4. Comment 4 (Full reviewer comment text reproduced): Statistical analysis of results: We do not have extensive experience in statistical analysis, but from a conceptual point of view, since, pneumonia, deep vein thrombosis and alcohol use were associated with delirium in univariate analysis (table 1), it would seem important to know whether albumin, neutrophils and other inflammatory indices are strongly associated with those categories and whether this has significant bearing on the conclusions of the study. For example, the only blood parameter associated with delirium in this cohort, NPAR has a p value of 0.037 while the association with Pneumonia reaches a much higher level of confidence (p=0.005). Both alcohol use and DVT are also significantly associated with delirium at a similar level of confidence to the association with NPAR. This needs explicit consideration in the paper.

Response:

Thank you for this very insightful question, which addresses the core of our statistical interpretation and the clinical relevance of our findings. We have revised our Discussion section to explicitly address these important points.

1. On Confounding Factors:

As you correctly suggest, it is crucial to distinguish independent predictors from confounding variables. Our multivariate logistic regression analysis (Table 4) was designed specifically for this purpose. This analysis confirmed that a high NPAR (≥22) is a strong and independent predictor of delirium, with an odds ratio of 9.14, even after adjusting for other baseline factors such as alcohol use.

Regarding pneumonia and DVT, we did not include them in the final predictive model because we classified them as concurrent or subsequent complications, rather than baseline predictors. Our study's primary objective was to identify risk factors present at the time of admission that could predict a future event (delirium). Since complications like pneumonia and DVT develop after admission, they are not suitable for an early predictive model focused on baseline risk stratification.

2. On the Strength of Association:

We also appreciate the opportunity to clarify our interpretation of the strength of the association. While a p-value indicates the statistical certainty that an effect exists (i.e., is not due to chance), the Odds Ratio (OR) is the appropriate measure of the magnitude or "strength" of that effect in a logistic regression model.

As noted, the OR for NPAR was 9.14, indicating a very strong clinical association: patients with a high NPAR on admission were over nine times more likely to develop delirium. The primary goal of our study was to identify an early, simple, and accessible biomarker. NPAR, calculated from a routine blood test upon admission, perfectly fits this role. This distinguishes it from later-onset complications like pneumonia, which are not available as predictive markers at the initial point of care.

To ensure these important points of interpretation are clear to our readers, we have added a comprehensive paragraph to our Discussion section clarifying the rationale for our predictive model and the interpretation of NPAR's value as an early biomarker.

We elaborated that NPAR remained an independent predictor in multivariate analysis (OR 9.14) even after adjusting for alcohol use. Pneumonia and DVT were excluded as post-admission complications, not baseline predictors. Discussion revised accordingly.

5. Comment 5 (Full reviewer comment text reproduced): Since there was no mention of an investigation into the relationship between alcohol use and delirium in the introduction or the methods sections of the manuscript nor any mention of why alcohol is singled out here when pneumonia appears to be a much stronger association, the authors should clarify in the methods section whether alcohol use was a pre-specified exposure variable or a secondary outcome of interest. If alcohol was included in the multivariate analysis (while pneumonia was not) this suggests that the authors have quantitative data on alcohol rather than simply categorical (as might be the case for pneumonia). If this is the case, then shouldn’t some continuous variable for alcohol be included (blood alcohol level?)

Response: Thank you for this important question regarding our analytical strategy and the selection of variables for the multivariate model.

First, regarding the inclusion of alcohol use: Alcohol is a well-established and significant risk factor for both traumatic injuries and the subsequent development of delirium. For this reason, it was pre-specified in our analysis plan as a key potential confounder that required adjustment in our model to isolate the effect of other variables like NPAR. The data for alcohol use was collected as a categorical variable (yes/no) based on interviews with the patient or their family upon admission; we did not have quantitative data such as blood alcohol levels. We have now clarified this data collection method in the Data collection subsection of our Methods.

Second, the decision to exclude pneumonia from the predictive model, despite its strong statistical association in the univariate analysis, was based on a crucial temporal relationship. Our analysis of patient timelines revealed that in our cohort, the onset of pneumonia always occurred after the onset of delirium. Because pneumonia was not a preceding event, it cannot be considered a predictor or a causative factor for the onset of delirium in this context. Instead, it should be viewed as a subsequent complication. Our model was specifically designed to identify baseline predictors available upon admission, making the inclusion of a post-onset complication methodologically inappropriate for our research question.

To address this point in conjunction with the related concerns about statistical interpretation raised in your previous question (Point 4), we have added a single, comprehensive paragraph to our Discussion section. This new text clarifies our complete rationale for variable selection in the multivariate model and reinforces our conclusion regarding the value of NPAR as an early predictive biomarker.

6. Comment 6 (Full reviewer comment text reproduced): The calculation of the cut off value for each analyte/ROC curve and how sensitivity and specificity values for the ROC curve were calculated might be explained a little more clearly. Is the value of 22 for NPAR in the normal range or is this actually a high NPAR value? Related to this, what is the normal range for NPAR (i.e in non-spinal cord injury individuals) and how does the raised NPAR value arise? i.e. the higher NPAR value could arise from raised neutrophil numbers or from lowered albumin levels. Knowing the absolute values for neutrophil numbers (per blood volume) and albumin concentration will provide insights that are useful to understanding the biological processes at play (and therefore useful to the delirium field).

Response:

Thank you for these invaluable suggestions, which have prompted us to perform additional analyses that have significantly deepened the interpretation of our findings. We have revised the manuscript extensively to address all three of your points.

1. Clarification of Methods: We have now specified in the Statistical Analysis subsection that the optimal ROC cutoff values were determined using the Youden index.

2. Clinical Context of NPAR: To contextualize our findings, we have added to the Discussion sec

---

## [Editor Report · Decision Letter 1]

2 Dec 2025

Association between delirium and neutrophil percentage-to-albumin ratio in patients with cervical spinal cord injury

:A single-center, retrospective study

PONE-D-25-32009R1

Dear Dr. Kumagai,

We’re pleased to inform you that your manuscript has been judged scientifically suitable for publication and will be formally accepted for publication once it meets all outstanding technical requirements.

Kind regards,

Pedro Kallas Curiati, M.D., Ph.D.

Academic Editor

PLOS ONE

---

## [Editor Report · Acceptance letter]

PONE-D-25-32009R1

PLOS One

Dear Dr. Kumagai,

I'm pleased to inform you that your manuscript has been deemed suitable for publication in PLOS One. Congratulations! Your manuscript is now being handed over to our production team.

Kind regards,

on behalf of

Dr. Pedro Kallas Curiati

Academic Editor

PLOS One